# A Dynamic Anomaly Detection Approach Based on Permutation Entropy for Predicting Aging-Related Failures

**DOI:** 10.3390/e22111225

**Published:** 2020-10-27

**Authors:** Shuguang Wang, Minyan Lu, Shiyi Kong, Jun Ai

**Affiliations:** School of Reliability and Systems Engineering, Beihang University, Beijing 100089, China; wsguang@buaa.edu.cn (S.W.); lmy@buaa.edu.cn (M.L.); buaaksy@buaa.edu.cn (S.K.)

**Keywords:** software aging, failure prediction, anomaly detection, machine learning

## Abstract

Software aging is a phenomenon referring to the performance degradation of a long-running software system. This phenomenon is an accumulative process during execution, which will gradually lead the system from a normal state to a failure-prone state. It is a crucial challenge for system reliability to predict the Aging-Related Failures (ARFs) accurately. In this paper, permutation entropy (PE) is modified to Multidimensional Multi-scale Permutation Entropy (MMPE) as a novel aging indicator to detect performance anomalies, since MMPE is sensitive to dynamic state changes. An experiment is set on the distributed database system Voldemort, and MMPE is calculated based on the collected performance metrics during execution. Finally, based on MMPE, a failure prediction model using the machine learning method to reveal the anomalies is presented, which can predict failures with high accuracy.

## 1. Introduction

In modern society, the complexity of software is continuously increasing, which brings convenience but also increases challenges to maintain software reliability. Rapidly developing cloud software based on distributed systems is a typical example and has attracted a lot of attention. It has two characteristics of high complexity and being long-running. Due to the accumulation of errors and garbage, it usually suffers from performance degradation or an increase in failure rate. This phenomenon is also called software aging [1].

Software aging has caused tremendous damage to many complex long-running systems such as web servers [2], operating systems [3], and even safety-critical software [4]. In a complex software system, aging is usually a dynamic nonlinear change process affected by many factors such as software errors, workloads, resource consumption, etc., which gradually cause the system to an error state and eventually to fail [5]. Memory-related types of aging are the most concerned in the existing research [6]. For example, a system failure caused by memory exhaustion due to memory leaks and memory fragmentation is a typical software aging-related failure [7,8]. As pointed out in the literature [1], it is the accumulation of aging-related errors that causes the internal environment of the system to enter a state where the aging-related errors are propagated, thus leading to aging-related failures (ARFs).

Software rejuvenation is an effective approach to delay or prevent the occurrence of ARFs [9]. Software rejuvenation can reconfigure the software by releasing resources, deleting garbage storage, etc., thereby significantly reducing performance degradation and failure rates caused by software aging [10]. The optimal schedule for software rejuvenation to perform is when it is close to the occurrence of a software failure [11]. Therefore, predicting the occurrence of software aging-related failures is essential to decide the optimal schedule for triggering *software rejuvenation*.

During the aging process, the system suffers from performance degradation before the failure occurrence. Thus, performance indicators can be used to predict ARFs. However, limited by the complexity of the system, effectively predicting failure is still a challenging task, especially because of the following three challenges:**Hidden failure-prone state:** Software aging is a gradual accumulation process. There is no prominent feature indicating the system is in a failure-prone state before ARFs, which is not only related to the current performance but also the previous performance indicators.**Fluctuating noises:** The system is in a dynamic and long-term running state, and it is inevitably interfered with by some fluctuating noises when collecting performance indicators. The presence of these noises may lead to misinterpretation of instantaneous anomalies as ARFs, leading to an increase in the predicted false alarm rate.**Multidimensional factors:** ARFs are the result of the simultaneous impact of multidimensional factors from the internal environment and external environment. Single-dimensional performance indicators or linear models are challenging to predict the dynamic and nonlinear software aging process.

Through the analysis of the challenges above, a novel approach for predicting the ARFs is proposed based on the combination of the dynamical time series analysis method and anomaly detection in machine learning. First, a new software aging indicator, Multidimensional Multi-scale Permutation Entropy (MMPE), is proposed, which is calculated on the time series of performance indicators. Permutation entropy (PE) is a complexity measure for time series analysis [12]. This method is used to extract nonlinear state information hidden in time series. It takes the advantages of simple calculation and can effectively detect dynamical changes in time-series [13]. Based on the analysis of the third challenge, PE is modified to an MMPE, which is appropriate for multidimensional and multi-scale time series. In this paper, MMPE is used to extract the dynamic changes of the system from the normal state to the failure-prone state with performance anomalies before the ARFs occur. Then the failure-prone anomalies are detected by the anomaly detection algorithm Isolates forest and One-class Support Vector Machine.

To deal with software aging, quite a lot of research has been developed for predicting the ARFs. The approaches proposed in previous studies are classified into model-based approaches and measurement-based approaches. The basic idea of model-based approaches is to provide state-based models that represent the degradation level of the system. Okamura et al. [14] combine the continuous-time Markov chain (CTMC) with system attributes distributions to propose a continuous-time Markov chain (CT-HMM) representing the degradation level of the system.

The measurement-based approaches usually monitor system variables and analyze the data collected during the runtime statistically. Our approach can be regarded as a measurement-based approach. The measurement-based approaches in previous studies mainly include time-series analysis, thresholding, and machine learning.

To predict the ARFs, the time-series analysis is used for analyzing the aging trend of software. The work [15] presents a stochastic time series decomposition algorithm based on robust locally weighted regression (Loess) to estimate the aging trend related to the exhaustion of system resources. J. Zhao et al. [16] introduce a method based on a non-stationary time series model to research the phenomenon of software aging. Araujo et al. [17] predict resource exhaustion time by combining time-series analysis with a threshold based on Memory Usage. J. Li et al. [18] adopt a hybrid approach including the heuristic-based threshold for predicting the ARFs.

P. Chen et al. [19] present an ARF-Predictor using entropy. They extend sample entropy to Multidimensional Multi-scale sample entropy as an aging indicator and develop ARF-Predictor containing threshold-based approaches namely FT and FT-X. Our approach differs from P. Chen et al.’s approach in terms of entropy and analysis. (1) We introduce the permutation entropy which calculates the permutation of the reconstructed sequence. P. Chen et al. introduce the sample entropy which measures the complexity of a time series via calculating the self-similarity of the sequence. (2) In our approach, the unsupervised anomaly algorithms are utilized to detect the failure-prone anomalies based on the aging indicators. P. Chen et al. use the monotonicity of the aging indicator MMSE to determine the threshold via threshold-based approaches.

Machine learning approaches infer a system state as normal or failure and identify failure-prone anomalies from algorithms such as classifiers and regressors. In the paper [20], K. Do et al. propose an energy-based model computed with an RBM to detect failure-prone anomalies. Y. Qiao et al. [21] apply the Long Short-Term Memory Neural Network(LSTM NN) to predict the software aging indicators including the system’s free physical memory and application’s heap memory.

The paper is organized as follows. The details of the proposed software aging indicator are elaborated in Section 2. Section 2 also demonstrates the procedures of the ARFs prediction approach, including the parameter selection for MMPE and the design of the signature window for the failure-prone anomalies. The experimental setup is described in Section 3 and the experimental results are analyzed in Section 4. Our work is concluded in Section 5.

## 2. ARFs Prediction Approach

The procedure of the ARFs prediction approach is presented in Figure 1. The ARFs prediction approach contains four modules: the data collection module, the feature selection module, the MMPE calculation module, and the failure prediction module. The performance indicators during the execution are monitored from different levels in the data collection module. Then, the principal component analysis (PCA) is applied to reduce the dimensionality of the performance indicators in the feature selection module. The MMPE values of the time series composed of the reduced performance indicators are calculated in the MMPE calculation module for the next step. The failure prediction module consists of the signature window and anomaly detection. The signature window is designed to label the failure-prone anomalies in the time series before the failure. Then the approach predicts the ARFs through detecting the persistent failure-prone anomalies before the ARFs occur.

The details of the ARFs prediction approach are introduced in this section, except for the data collection module, which will be presented in the experiment setup.

### 2.1. Software Aging Indicator Based on MMPE

In complex software systems, the performance degradation exhibits nonlinear and dynamic manner during the software aging. Thus, the complexity measure of analyzing time series is introduced as a novel software aging indicator to detect dynamical changes in the software system. The concept of entropy has been an essential measure of the complexity of time series generated from nonlinear dynamical systems. Bandt and Pompe introduced PE as an appropriate method that maps the continuous time-series to a symbolic sequence [12]. PE uses the phase space reconstruction method to delay the coordinate state space, analyzes a one-dimensional time series in a nonlinear system, and mines the hidden characteristics existing in the system [13].

The specific details of modifying PE to the software aging indicator MMPE can be divided into three parts. The basic principle of permutation entropy is introduced in the first part. In the second part, the coarse-grained process is applied to modify PE to MPE for multi-scale time series. In the third part, MPE is modified to MMPE, which is appropriate for multidimensional and multi-scale time series. Then MMPE is used as a software aging indicator to reveal the dynamic transition of the system from a normal state to a failure-prone state.

#### 2.1.1. Permutation Entropy Algorithm

The initial object of the permutation entropy (PE) algorithm is a one-dimensional time series $\left\{x\left(i\right),i=1,2,\mathrm{...},N\right\}$. To extract more useful information from a one-dimensional time series, the first step is to partition it into a matrix of overlapping vectors by the phase space reconstruction method according to the Takens theorem [22]. This step contains two hyperparameters: the embedding dimension m and the embedding time delay L. In the obtained reconstructed m-dimensional matrix:$X\left(i\right)=[x\left(i\right),x\left(i+L\right),\mathrm{...},x\left(i+\left(m-1\right)L\right]$, m controls the length of the new vectors. The L controls the interval between elements of each of the vectors, where $1\leq i\leq N-\left(m-1\right)L$.

After the phase space reconstruction, the m-dimensional matrix is uniquely mapped into the permutations according to the ordinal rankings. The given $X\left(i\right)=[x\left(i\right),x\left(i+L\right),\mathrm{...},x\left(i+\left(m-1\right)L\right]$ are rearranged in ascending order as $X\left(i\right)=\left[x\left(i+\left(j_{1}-1\right)L\right)\leq x\left(i+\left(j_{2}-1\right)L\right)\leq \mathrm{...}\leq x\left(i+\left(j_{m}-1\right)L\right)\right]$. In the rearranged sequence, $j_{1},j_{2},\mathrm{...},j_{m}$ denote the location of the elements in $X\left(i\right)$. It is worth noting that when an equivalence such as $x\left(i+\left(j_{i1}-1\right)L\right)=x\left(i+\left(j_{i2}-1\right)L\right)$ appears, the order is arranged according to the value of the subscribe j. If $j_{i1}<j_{i2},$ then x is sorted as $x\left(i+\left(j_{i1}-1\right)L\right)\leq x\left(i+\left(j_{i2}-1\right)L\right).$ Therefore, any vector $X\left(i\right)$ can be uniquely mapped into a group of subscript sequences as $\left\{j_{1},j_{2},\mathrm{...},j_{m}\right\}$. If each symbol indexed by i is different, the m-dimensional embedding matrix has at most m! permutations. Then the probability of each permutation $(P_{1},P_{2},\mathrm{...},P_{k}$, where k≤m! ) can be calculated by counting the times of the permutation in the total sequences. Finally, the PE is defined in the form of Shannon entropy as:(1)$$H_{p}\left(m,L,N\right)\;=\;-\overset{k}{\underset{j\;=\;1}{\sum }}P_{j}lnP_{j}$$
when $P_{j}\;=\;1/m!$, then $H_{p}\left(m,L,N\right)$ attains the maximum value ln(m!). For convenience, $H_{p}\left(m,L,N\right)$ is normalized by ln(m!) as in the following:(2)$$0\leq H_{p}\left(m,L,N\right)=\;H_{p}\left(m,L,N\right)/\ln(m!)\leq 1$$

From the derivation process of $H_{p}\left(m,L,N\right)$, it can be seen that PE is defined to measure the complexity and randomness of the time series. The lower the values of $H_{p}\left(m,L,N\right)$ are, the more steady the time series is.

#### 2.1.2. Multi-Scale Permutation Entropy Algorithm

To express the structural complexity at different scales, PE is modified to Multi-scale Permutation Entropy (MPE) to contain more hidden characters. MPE is calculated by PE of the coarse-grained time series over different scales [23]. For $X\;=\;\left\{x\left(i\right),\;i\;=\;1,2,\ldots ,N\right\}$, a coarse-grained sequence $Y^{\left(\tau \right)}\;=\;\{y_{j}^{\left(\tau \right)}|j\;=\;1,2,\mathrm{...},\left[\frac{N}{\tau }\right]\}$ is constructed as the following steps: First, the original time series is split into continuous and non-overlapping windows of the size τ. Second, the data are averaged inside each window. Finally, the formula of $y_{j}^{\left(\tau \right)}$ is defined as:(3)$$y_{j}^{\left(\tau \right)}\;=\;\frac{1}{\tau }\overset{j\tau }{\underset{i\;=\;1+\left(j-1\right)\tau }{\sum }}x_{i}\;1\leq j\leq \left[\frac{N}{\tau }\right]$$

The τ  is the scale factor and $\left[\frac{N}{\tau }\right]$ denotes the rounding integer digit of $\frac{N}{\tau }$. An example of the coarse-grained process with the scale factor τ = 2 is shown in Figure 2.

The process of the coarse-grained sequence corresponding to each scale in the range of [1,τ] is calculated separately. After that, we get a set of permutation entropy {$H_{p}^{\left(1\right)}\left(m,L,Y\right),\;H_{p}^{\left(2\right)}\left(m,L,Y\right),\ldots ,H_{p}^{\left(\tau \right)}\left(m,L,Y\right)$}, and each element corresponds to the PE of a scale. The mean value of τ permutation entropy is used to represent the multi-scale entropy (MPE) with scale factor τ.

#### 2.1.3. Multidimensional Multi-Scale Permutation Entropy Algorithm

The MPE introduced in the previous section is appropriate for single dimensional time series. However, the performance indicators collected during the execution contains multidimensional variables; MPE cannot be directly used as an aging indicator for multidimensional time series. Thus, the MPE is extended to MMPE through the following steps.

The first step is the normalization of performance metrics. The multidimensional performance metrics have multiple ranges and units. The significant changes in the numerical fields between performance indicators with different units (such as CPU utilization and the total memory utilization) may affect the accuracy of MMPE. To remove the unit limits for performance indicators, the performance indicators are normalized and converted to dimensionless values. For the given d-dimensional matrix: $X\;=\;\left[X_{1},X_{2},\ldots X_{d}\right],\;$ each column denotes the time series of a particular performance indicator, and the length of each time series is *N*. Then X is normalized as in the subsequent step:(4)$$X_{ji}^{\prime }\;=\;\frac{X_{ji}-\min(X_{i})}{\max(X_{i})-\min(X_{i})},1\leq i\leq d,1\leq j\leq N$$

The MPE of the i−th column is $\mathrm{MPE}\left(m_{i},L,Y\right)$, where the $m_{i}$ denotes the embedded dimension of the i−th column. For the normalized matrix $X^{\prime }$, a vector of MPE is obtained as $MPE\;=\;\left[MPE\left(1\right),MPE\left(i\right),\mathrm{...},MPE\left(d\right)\right]$. To get an integrated entropy as a system indicator, the 2-Norm is introduced in our approach for calculating the MMPE as the following step:(5)$$MMPE\;\;=\sqrt{\overset{d}{\underset{i\;=\;1}{\sum }}MPE{\left(i\right)}^{2}}$$

The 2-Norm (also known as the Euclidean norm) is often used to calculate the length of the vector [24]. MPE values of an ideal system state without any changes is a zero vector, and MMPE can represent the Euclidean distance from an MPE vector to the zero vector. Consequently, MMPE can be used as a software aging indicator to measure the deviation of the system state dynamically. The pseudo-code of MMPE is presented in Algorithm 1 to clearly describe the whole process.

Algorithm 1 The algorithm of the Multidimensional Multi-scale Permutation Entropy (MMPE) Calculation Procedure.

**Algorithm 1.** MMPE Calculation Procedure

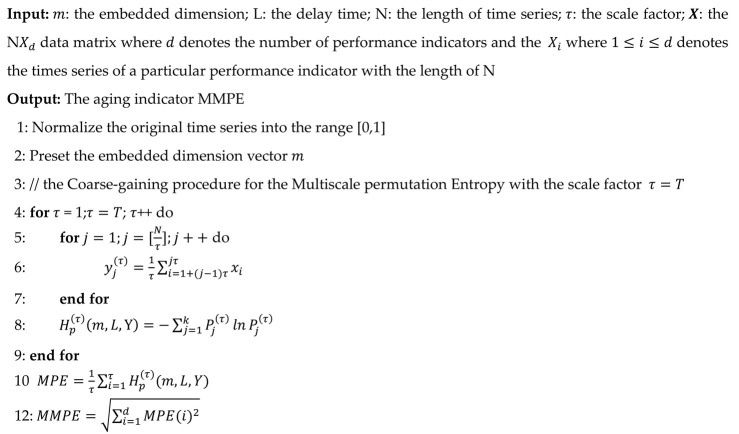



### 2.2. MMPE Calculation

During the software aging process, the multiple time-varying performance indicators are monitored as a multivariate time series. Multivariate time series provide rich information for detecting performance anomalies, but also increase the workload of calculation to a certain extent. More importantly, there may be correlations between these performance indicators, thereby increasing the complexity of feature analysis. If each indicator is analyzed in isolation, the information in the data cannot be fully utilized. Therefore, the principal component analysis (PCA) method is used to reconstruct a set of linearly uncorrelated subsets based on the original high-dimensional performance indicators [25]. The high-dimensional performance metrics are reduced to 10-dimensional performance metrics through principal component analysis, and the information loss is negligible. Then the MMPE values of the time series constructed by reduced performance indicators are calculated for anomaly detection.

Our method for detecting performance anomalies is based on the MMPE algorithm combined with the sliding window. A window of length N is designed to slide on the time series and divide the long time series composed of performance indicators into overlapping blocks. Each overlapping block corresponds to a data subset of length N, and MMPE of each data subset is calculated for detecting performance anomalies, as shown in Figure 3 the time step of the sliding window is set to 1.

The changes in the MMPE values with time are excepted to accurately indicate the dynamic transformation of the system state during the software aging process. Therefore, before figuring MMPE, it is necessary to consider how to choose the appropriate embedding dimension m, the delay time L, the data length (window size) N , and the scale factor τ.

In the coarse-grained process for different time series scales, the window size N and the scale factor τ need to be considered together. The principle of the value of the data length N and the scale factor τ is that the coarse-grained sequence cannot affect the calculation of MMPE. Our purpose is to accurately detect the performance anomalies of the system before the ARFs occurring, so the sliding window should not be set too large [26]. However, the window size should not be too small; otherwise, MMPE will have a significant deviation. By setting N = 500, 1000, and 2000 to compute PE, it is can be found that the results are similar. Therefore, combined with the actual data set length, N is taken as 1000 in our approach. After fixing the size of the time window, the range of scale factor τ is determined as 1~10.

When reconstructing the phase space, if the embedding dimension m is too small, the reconstructed time series contains fewer features than the original series, which will cause information loss. If the value of m is too large, the sensitivity of the algorithm to detect performance changes will decrease, and it will also increase the complexity of the calculation. Bandt and Pompe suggested in their paper that m takes a value between three and seven [12]; To compare the influence of parameters on MMPE, we take 20 sets of different subsequences from the collated data set to calculate MMPE. The data length is 1000, the delay time L is 1 and the scale factor increases from 1 to 10. The embedding dimension  m is taken in {3,4,5,6,7}. The average values of those MMPEs are shown in Figure 4a. The error bars corresponding to the standard deviations are also shown in Figure 4a. As shown in Figure 4a when  m is small (for  = 3, 4 ), the values of MMPE have no significant changes in overall scale factors. If m is too large, minor changes in time series will be ignored. Moreover, the values of MMPE are calculated 100 times separately and record the total consumption times for each embedding dimension. Figure 4b shows that the larger the embedding dimension, the longer the calculation time. When  m = 7, calculating one hundred MMPEs consumes a total of 8.82 s. To predict ARFs in time, the calculation is expected to consume as little time as possible. Thus, m is set as 5 according to the above analysis. Additionally, the delay time L is set to 1 as Bandt and Pompe have done in their original work [12].

### 2.3. Failure-Prone Anomalies Detection

Basing on the aging indicator MMPE, the failure-prone anomalies are detected to predict upcoming ARFs in time. Thus, the signature window and anomaly detection techniques are combined to achieve failure prediction. In anomaly detection algorithms, anomalies are also considered outliers and deviations to be detected [27]. This concept is consistent with the proposed software aging indicator MMPE, which indicates the deviation of the system from the normal state [28]. Several anomaly detection techniques have been proposed in the literature, such as density-based techniques, and cluster analysis-based outlier detection [29,30]. The most popular anomaly detection algorithms, Isolated Forest [31] and One-Class Support Vector Machine (OC-SVM) are selected as two different anomaly detection models in our approach.

#### 2.3.1. Signature Window for Failure-Prone Anomalies

The software aging process is a process in which performance gradually decreases with time. The system is in a failure-prone state with performance anomalies before the ARFs occur [28]. Therefore, a signature window with the size of l is designed to label the soft aging indicators caught in the window as failure-prone anomalies. As shown in Figure 5, the right border of the signature window is the point of failure occurring, and the timestamps within the window correspond to the failure-prone anomalies. The first occurrence of anomalies in the signature window is accompanied by the system beginning to exhibit anomalous conditions, which continue to appear in the window until the ARFs occur [32]. This approach can avoid false positives. For example, some performance anomalies are related to high pressure in a short time but will not cause software aging-related failures. Failure signatures for anomalies can distinguish between failure-prone behavior and normal state, so it is particularly effective in predicting failures [33]. When the signature window includes only a small portion of failure-prone anomalies, the prediction results may be inaccurate. In the subsequent experiments, the signature windows with different sizes will be set to compare the impact of window size on the prediction results and identify a suitable window size.

#### 2.3.2. Anomaly Detection Method

To avoid false positives and improve the accuracy of failure prediction, an approach which combines the MMPE algorithm and the anomaly detection algorithm is proposed. First, the monitored performance indicators during execution are used as the input of the MMPE algorithm and obtain MMPE that measures the complexity of the system as a novel software aging indicator. The signature window of length l is designed to label the failure-prone anomalies before ARF occurs. Due to the ARFs occurring after the software system has been running for a long time, the normal state lasts much longer than the failure-prone state. Correspondingly, the samples collected in the normal state are far larger than the anomaly samples which are collected in the failure-prone state. Thus, the ARFs dataset has a serious class imbalance problem. This condition will negatively affect the standard classifier applied to the class balance situation, and also affect the efficiency of classification. Therefore, to reduce the impact of the serious imbalance between the two types of samples, the anomaly detection algorithms are selected from the classification models as the prediction model.

In this section, the Isolation Forest and OC-SVM as two different anomaly algorithms are used to choose the appropriate window size in the dataset collected through the experiment. At the same time, they have further verified the applicability of our method on the public data set. Isolation Forest and OC-SVM are the unsupervised algorithms used to detect anomalies. Unsupervised anomaly detection technology assumes that most of the samples are normal, and detects anomalies in the unlabeled test data set by finding the most outlier instances [34,35].

1. Isolated Forest anomaly detection method

The principle of Isolated Forest is to isolate anomalies instead of the most commonly used technique for analyzing regular points [36]. Due to anomalies accounting for a small percentage of the data set and deviating greatly from normal, they are more likely to be “isolated” from normal values. The isolated Forest algorithm builds a binary search tree as an isolation tree by using multiple iterations without the need to define a parameter model. The anomalies are the points on the isolation tree that have shorter average path lengths. The abnormal point detection process can be described as three steps:

X samples of the train set are randomly selected from the training set to construct an “iTree”. Then the steps are iteratively repeated one according to the sample data capacity to create the “iTrees” to construct a binary tree forest.The isolated Forest algorithm uses the expected average length of the binary search tree to estimate the average of “iTrees” length. For the given set of samples of size n, the expected length is:(6)$$c\left(n\right)=\;ln(n-1)-\frac{2\left(n-1\right)}{n}+c_{o}$$$c_{0}\;$ is a constant and $c\left(n\right)$ is the expected value of the path length which is used to normalize each path length.The binary search tree is recursively traversed and the path length $h\left(x\right)$ is recorded from the root node to the leaf node. The expected value $E\left(h\left(x\right)\right)$ of all data samples is calculated by using statistical methods and then evaluate the abnormal points that deviate from the normal range. The abnormal score S is calculated as the following
(7)$$s\left(x,n\right)\;=\;2^{-\frac{E\left(h\left(x\right)\right)}{c\left(n\right)}}$$

2. OC-SVM anomaly detection method

The OC-SVM is an unsupervised outlier detection introduced by B. Schölkopf [37], which estimates the support for high-dimensional distribution. For the given training dataset $D\;=\;\left\{x_{i},\;i\;=\;1,\;2\;,\mathrm{...},N\right\}$, where $x_{i}$ is a n -dimensional vector. The samples in the training data that are far away from other samples are considered as outliers. Therefore, the outlier detection estimator focuses on fitting the most concentrated areas in the training dataset and can be used as the anomaly detection for the abnormal samples. The margin of the OC-SVM corresponds to the probability of finding a new but regular observation outside the boundary defined by the kernel and scalar parameter. The OC-SVM uses the kernel function to map the original low-dimensional feature space $R^{n}$ to the high-dimensional space χ, the corresponding nonlinear mapping is $\phi \left(x_{i}\right)$. Then a hyperplane represented by the support vector is obtained through the sample training, which can separate the abnormal sample from the origin as much as possible. The hyperplane of the high space is defined as:(8)$$w\cdot \;\phi \left(x_{i}\right)-\rho \;=0$$
where w is the normal vector of the hyperplane and *ρ* is the intercept of it. In the case of correctly distinguishing as many target samples as possible, the distance from the origin to the hyperplane is maximized. The optimization of the OC-SVM is the following:(9)$$\underset{w,b,\xi_{i}}{\min}\left(\frac{1}{2}{\mathrm{||w||}}^{2}+\frac{1}{vn}\overset{n}{\underset{i\;=\;1}{\sum }}\xi_{i}-\rho \right)$$
(10)$$s.t.\;\left(w\cdot \phi \left(x_{i}\right)\right)\geq \rho -\xi_{i}\;\mathrm{and}\;\xi_{i}\geq 0\forall i$$
where $\xi_{i}\geq 0$ is a slack variable and $v\;\epsilon \;\left(0,1\right]\;$ is a predefined parameter, which means the lower bound percentage of the support vector. To solve the above optimization problem, the Lagrange function is constructed and then the classification decision function is obtained as the following: (11)$$f\left(x\right)\;=\mathrm{sgn}(\overset{n}{\underset{i\;=\;1}{\sum }}\alpha_{i}Κ\left(x_{i},x\right)-\rho )$$

$Κ\left(x_{i},x\right)$ is the kernel function and $\alpha_{i}$ is the Lagrange factor; sgn(x) presents the symbolic function, if x>0, the test sample is determined to be an abnormal sample, otherwise it is determined to be a normal sample.

The anomalies detected by the above anomaly detection algorithms correspond to the abnormal state of the system. The ARFs are predicted by the detection of continuous failure-prone anomalies.

## 3. Experimental Verification

To verify the applicability of the prediction approach for the software aging-related failure problem in a real situation, the approach was tested via the case study based on Project Voldemort, which is used at LinkedIn. First of all, we introduce the experiment platform and the experiment setup for proving our ARFs prediction approach in this section. Then, different workloads are designed to simulate the software aging process and monitor the key performance indicators during the running time for detecting the failure-prone anomalies.

### 3.1. Case Studies

To explore the effectiveness of our approach for predicting ARFs, we propose the following two research questions:

RQ1: Does the size of the signature window impact on the effectiveness of the prediction approach?

We set up a set of experiments with different window sizes on the same training set which contains the MMPE and original performance indicators. The suitable size of the signature window was determined by comparing the average effectiveness of the anomaly detection models.

RQ2: Does the new software aging indicator MMPE effectively improve the accuracy of predicting failure? Is its effect universal?

We set up two sets of experiments, using the original performance indicators and the performance indicators with MMPE as two different training sets. By comparing the average results of the prediction, the effectiveness of MMPE was analyzed. To verify its universality, we not only collected performance datasets through the designed experiment, but also selected public data sets for verification. The experiment designed for the ARFs prediction approach is introduced in the following section.

To verify the above two problems experimentally, an open-source distributed key-value storage and serving system, Project Voldemort^1^ was deployed as our test platform. Project Voldemort is the not only SQL (NoSQL) system that provides flexible storage used at LinkedIn by abundant critical services powering a large portion of the site. We set up the database system Project Voldemort 1.10.26 on a single node cluster with two data partitions. We used Yahoo! Cloud Serving Benchmark (YCSB^2^), an open-source framework for benchmarking cloud data serving [38], to generate the real-world workloads. All of the tests were run on a virtual machine VMware Workstation 15 Player configured with one vCPU, 3GB of RAM, and 20GB hard disk space, and runs the Ubuntu 18.04.3 LTS amd64 operating system.

### 3.2. Workloads Generation Module

To collect the performance indicators during software aging, workloads should be generated to simulate real applications and accelerate the process of software aging-related failures. In our experiment, the YCSB was applied to generate the workloads corresponding to real-world application situations. The workload generated by the YCSB client consists of two parts: the data set of records to be loaded, and the transaction set that defines the operations performed on the data set. The YCSB client will try to perform as many operations as it can possibly perform in each execution. For example, if each operation takes an average of 10 milliseconds, the client will perform about 100 operations per second per worker thread. Besides, these operations are subject to Zipfian distribution, which ensures that specific keys are accessed more frequently than other keys, thus simulating the general access mode of most websites.

In this paper, we keep the same data set and change the workload parameters for the transaction set to design two types of workload patterns, such as the following:*Update-workload:* This workload contains operations containing a mix of 50/50 reads and writes, the corresponding application example is a session store recording recent actions. In this workload pattern, the YCSB Client uses a single worker thread, and the Voldemort server is in a normal operational condition.*Write-workload:* This workload contains operations containing a mix of 75/25 writes and reads. We change the write rate by changing the time interval (called ***sleep_time***) between two consecutive requests generated by the same client. The corresponding application example is that users continually update their information.

The Update-workload is used to simulate the normal operation, and the Write-workload is used to accelerate the process of software aging-related failures through consuming the resources continuously. We first accelerate the process of software aging and then change the Write-workload pattern to the Update-workload pattern for the normal state after the failure. If the failure persists, it can be ruled out that it is a failure caused by the workload overload. It is worth noting that the maximum number of client threads for Voldemort is 100. The workloads of our application are within the configured range of the server. (1. Project Voldemort: http://www.project-voldemort.com/voldemort/; 2. YCSB: https://github.com/brianfrankcooper/YCSB)

### 3.3. Collection of Performance Indicators

To be able to sensitively detect the abnormal state of the software before aging and failure, we monitor the key performance indicators during software execution. We use Linux’s system performance analysis tool SAR (System Activity Report) to collect system activity information from multiple aspects, including CPU utilization, memory usage, disk I/O, network traffic, etc. Since Project Voldemort is written in Java and runs on the JVM, we also use java memory monitoring tool ***Jmap*** to collect heap memory usage information.

To avoid biases related to the workload patterns, we conduct a total of 20 groups of experiments by changing the parameters of the workloads. According to the workload patterns described in the previous section, we apply the ***Update-workload pattern*** for the normal operations to collect the performance metrics of the normal state, and the ***Write-workload*** is applied to accelerate the aging process for aging-related failures caused by the internal software errors (Aging-Related bugs). During the experiments, we collect performance indicators every five seconds, and the experiments will stop when the system crashes, or resources are exhausted. In our experiments, we observe two typical phenomena caused by aging-related failures:

***Server process killed***: When system memory exhaustion occurs, the server process of Voldemort will be killed by ***Out of Memory (OOM) kille****r*. This situation causes the system to panic and stop; the server cannot be restarted unless a software rejuvenation strategy is initiated. Figure 6 shows the trend of memory utilization before the ARFs occur; it can be observed that when the memory consumption exceeded nearly 95%, the failure occurred.

***Client connection timeout:*** Affected by software aging, the server cannot completely respond to the YCSB clients’ requests. The timeout error rate increases with the aging process.

Significant performance degradation can also prove the existence of software aging. The performance of Voldemort as a NoSQL system is measured by the average latency of the requests and the throughput. The average latency refers to the delay from the time the user sends the request to the response from the server. This is an indicator that can directly measure whether the software can meet user needs. As shown in Figure 7a, the average latency keeps increasing over time, which means that the performance of Voldemort is decreasing gradually. The throughput is the number of requests that the server can handle in a specific time unit. The larger the throughput, the greater the volume of business processed by the system per unit time, which directly reflects the business processing capacity and carrying capacity of the software. Figure 7b shows the decreasing trend of the throughput. Voldemort cannot recover performance after software aging failure occurs.

### 3.4. Data Processing

***Voldemort System Dataset:*** To eliminate the impact of different units, we normalized the performance indicators to obtain the dimensionless performance metrics. Then, we used the PCA method to convert the performance metrics that may be related to a new set of linearly uncorrelated synthesis through orthogonal transformation [39]. The performance indicators collected in each set of experiments are treated as a set of multidimensional time series. Thus, we first calculated the MMPE time series of each set of performance indicators time series separately and then aggregated them to a complete dataset. To verify the validity of the signature window, we took three different signature window lengths to label the failure-prone anomalies according to the index of timestamps before the ARFs occurred.

***Google Cloud trace Dataset (GCD):*** In addition to the datasets collected in the above experiment, we further applied our approach to the public dataset. However, few public datasets can be directly used for the ARFs prediction. This dataset was filtered from the Google Cloud trace Dataset (GCD) which is very representative in cloud computing. The GCD contains the CPU and memory utilization by summarizing their CPU and Memory utilization every five minutes for the tasks of each job. In this paper, four subsets were selected where the CPU utilization exhibited an increasing trend and the threshold exceeded 80%. Due to their performance being close to the software aging threshold, the corresponding failure reports in the data set are regarded as software aging failures for the ARFs prediction. A similar approach is verified in the references [40].

To standardize the number of groups in test sets and avoid overfitting, we apply the five-fold Stratified Shuffle Split to split the datasets. This cross-validation returns stratified randomized folds, which preserve the percentage of samples for each class [41]. The proportion of the test set is 30%, and the proportion of the train set is 70% in the dataset.

The prediction of the ARFs is transformed into a binary classification problem in our approach. Since the failure-prone anomalies account for a small percentage of the total samples during software aging running, there is a class imbalance. Thus, the smote method was applied to increase the proportion of anomaly samples to 30%. The confusion matrix was used for representing the four prediction results of the model for the positive class. The positive classes correspond to the predicted conditions that are failure-prone, where the True Positive (TP) presents the correct alarm, and the False Positive (FP) presents the false alarm. The negative classes correspond to the predicted conditions that are not prone to the ARFs, where the False Negative (FN) presents the missed alarm and the True Negative (TN) presents the correct normal alarm.

We used three metrics to evaluate the validity of the ARFs prediction results for positive classes. The ratio TP/(TP + FP) represents the precision, where TP is the number of true positives and FP the number of false positives. Intuitively, the precision represents the ability of the predictor not to mark negative samples as positive. This measure can assess the rate of false alarms in which the ARFs predictor misjudges normal conditions as failures [42].

The recall is the ratio TP/(TP + FN). This measure is the ability of the predictor to find all the positive samples, and thus can assess the percentage of failures that can be predicted.

The prediction models are expected to perform well in both precision and recall. However, it is hard to take care of the precision and recall in some conditions. For example, if the model expects to predict positive samples as much as possible, it will have a higher recall but a lower precision. If the model is conservative and only predicts positive samples with a high degree of confidence, then its precision will be relatively high, and the recall will be relatively low. Therefore, the f1-score is proposed as an indicator that comprehensively considers recall and precision. The f1-score is defined as the weighted average of precision and recall, and the relative contribution of them to the f1-score is equal. The f1-score reaches its best value at 1 and the worst score at 0. The formula for the f1-score is: f1-score = 2 * (precision * recall)/(precision + recall)(12)

The f1-score aims to improve precision and recall as much as possible while training the model.

The accuracy score is the ratio (TP + TN)/TP + TN + FP + FN, which presents the ratio of the number of correctly predicted samples to the total number of predicted samples.

## 4. Discussion of the Experimental Results

The effectiveness of our approach for predicting ARFs is verified in this section. First, the effect of different signature window sizes is analyzed by the comparison of prediction results. The most suitable signature window size is determined according to the best prediction results. Then the prediction results obtained by the ARFs prediction approach based on MMPE and original performance time series are compared to verify that MMSE can effectively improve the accuracy of predicting failure

### 4.1. Verify the Effectiveness of the Signature Window for Failure-prone Anomalies

To answer RQ1, the experiments are designed with three window sizes on the same training set. Because the time step of monitored indicators is 5 seconds, we set l = 15, 30, and 45 seconds as the lengths of signature windows. The anomaly detection models OC-SVM and Isolated Forest are trained on the Voldemort dataset. Then we compare the average effect of the three window sizes on the prediction results by the four metrics.

Table 1 contains the comparison results of the prediction effectiveness of OC-SVM and Isolated Forest with different signature window sizes. The overall prediction effectiveness of Isolated Forest performs better than that of OC-SVM. It can be seen that the two models with the signature window size of 30 seconds (N30) both show the best predictive effect. The recall of the Isolated Forest is 94% and the f1-score is 91%. It can be further verified by comparing the average effectiveness of the two prediction models that N30 is the most suitable signature window size. (3. Google Cloud trace Dataset: http://github.com/google/cluster-data).

It can be observed that the suitable window size has an obvious positive impact on the prediction results. When the window size is 45 seconds, it corresponds to the worst case. The precision of the OC-SVM is 0.72 and the f1-score is 0.71 in this case. The window size of 45 seconds is too large to reduce the sensitivity of the anomaly detection algorithm to the failure-prone anomalies. When the window size is 30 seconds, the prediction results of all aspects of the two anomaly detection algorithms are the best case. The recall of the Isolated Forest is 0.94 and the f1-score is 0.91 when the window size is 30 seconds. As for the OC-SVM, f1-score is 0.86. Therefore, the window size of 30 is chosen as the optimal choice for the next experiment.

### 4.2. Verify the Effectiveness and Universality of MMPE

Table 2 and Table 3 compare the prediction results of MMPE and Origin indicators on the Voldemort dataset and GCD dataset. The samples within 30 seconds before the ARFs are labeled as failure-prone anomalies and detected by the anomaly detection algorithm. MMPE presents the performance indicators including MMPE, and Origin presents the performance indicators without MMPE. The datasets that contain MMPE achieve better prediction effectiveness than the original performance indicators. It can be observed that the performance indicators including MMPE, significantly improve the efficiency of the prediction. As for the Voldemort dataset in Table 3, the f1-score of the OC-SVM increases from 0.72 to 0.86. Meanwhile, the recall of the Isolated Forest algorithm has a 13.2% increase and the f1-score increases from 0.86 to 0.91, which means that the comprehensive performance of recall and precision has been improved with the addition of the MMPE. As for the GCD dataset in Table 3, the addition of the MMPE also improves the prediction efficiency of the OC-SVM and Isolated Forest. The f1-score of the OC-SVM increases from 0.73 to 0.85, and it increases from 0.84 to 0.93 for the Isolation Forest. It can also be observed that the Isolated Forest performs better than the OC-SVM on both Voldemort Dataset and GCD dataset.

The increase in prediction efficiency is due to the that MMPE can reveal hidden system states. Compared with the original performance indicators, the MMPE algorithm reconstructs the time series, including the information in the time dimension. The system state revealed by MMPE not only corresponds to the current performance, but also to the previous values of the performance indicators. The outstanding performance of the ARF approach on the Project Voldemort and GCD dataset verifies its applicability for ARFs in complex software systems.

## 5. Conclusions

In this paper, an approach for predicting ARFs is proposed based on the dynamical anomaly detection method. We first propose a novel software aging indicator MMPE, calculated on the time series of performance indicators. To our knowledge, we are the first to apply the complexity measure permutation entropy (PE) of a dynamical system to predict the ARFs. Based on the aging indicators, a signature window is designed to label the failure-prone anomalies and use an unsupervised anomaly algorithm Isolation Forest to detect failure-prone anomalies. To verify the applicability of the prediction approach for the ARFs in a real situation, the software aging experiments run on Project Voldemort, a distributed key-value storage system that is used at LinkedIn by numerous critical services powering a large portion of the site. The experimental evaluation results in the distributed database verify that our approach for predicting the ARFs can achieve high accuracy. Compared to the original performance indicators, the addition of MMPE increases the f1-score of the Isolated Forest anomaly detection model from 0.86 to 0.91 on the Voldemort Dateset. Our approach provides a new effective way for the online prediction of the ARFs in complex software systems.

## Figures and Tables

**Figure 1 entropy-22-01225-f001:**
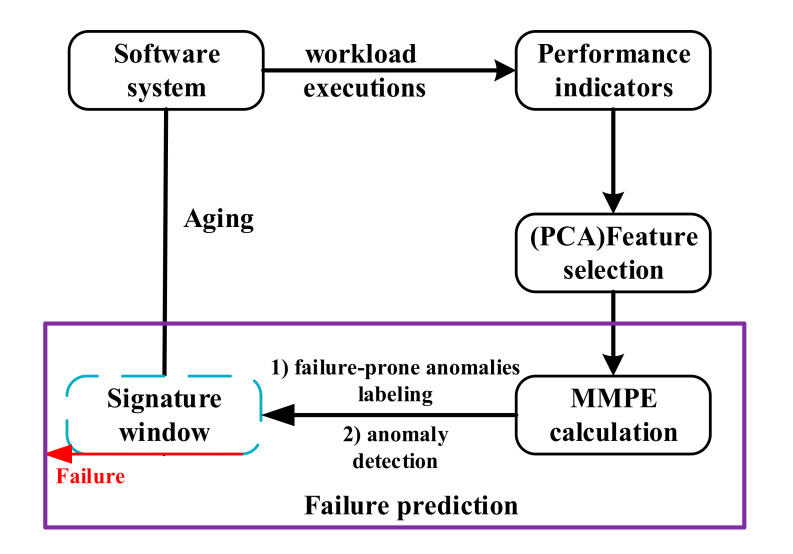
The process of the Aging-Related Failures (ARFs) prediction approach.

**Figure 2 entropy-22-01225-f002:**
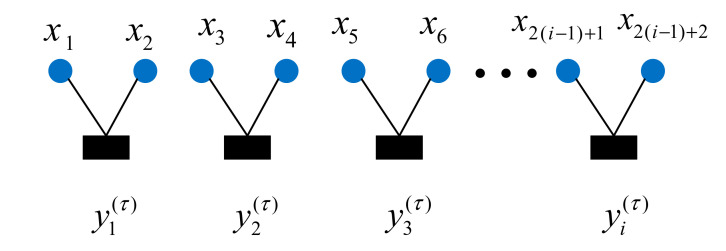
The coarse-grained process of Multi-scale Permutation Entropy (MPE) with the scale factor τ = 2

**Figure 3 entropy-22-01225-f003:**
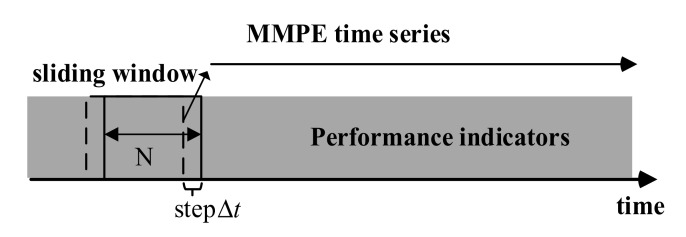
The calculation process of MMPE.

**Figure 4 entropy-22-01225-f004:**
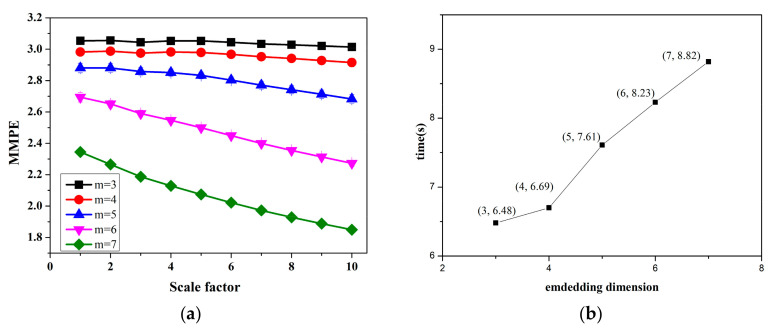
The comparative analysis of MMPE under different embedding dimensions: (**a**) The Comparison of MMPE values under different embedding dimensions (**b**) The consumption time of MMPE for different embedding dimensions.

**Figure 5 entropy-22-01225-f005:**
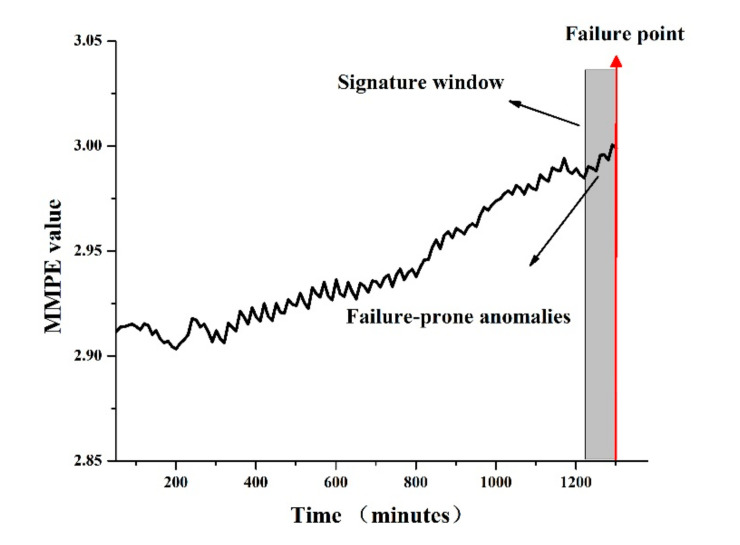
The signature window for failure-prone anomalies.

**Figure 6 entropy-22-01225-f006:**
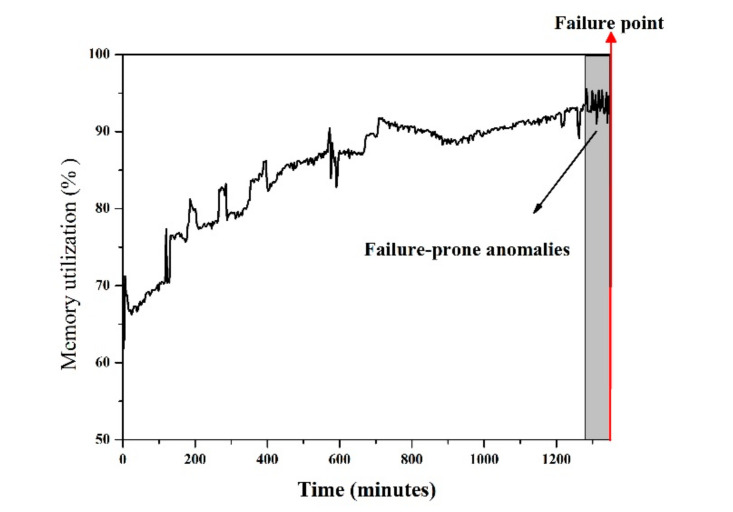
The increase of Memory utilization of the Voldemort for ARFs.

**Figure 7 entropy-22-01225-f007:**
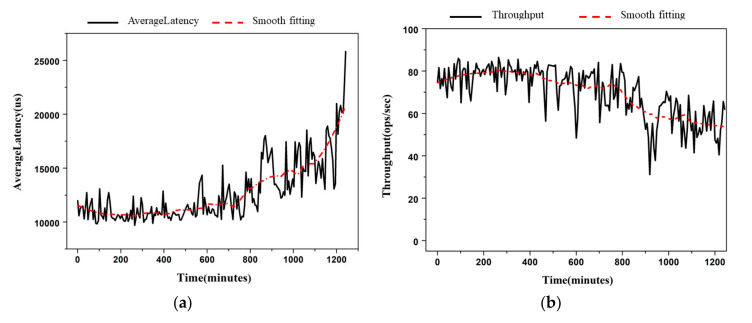
Performance degradation of the Voldemort for ARFs (**a**) the average latency of the requests (**b**) the throughput.

**Table 1 entropy-22-01225-t001:** The prediction results of the prediction approach with different anomaly detection algorithms and window sizes

Anomaly Detection Algorithms	Window Size	Precision	Recall	f1-Score	Accuracy
**OC-SVM**	15	0.73	0.75	0.74	0.82
	30	0.86	0.86	0.86	0.82
	45	0.72	0.71	0.71	0.80
**Isolated Forest**	15	0.84	0.92	0.88	0.97
	30	0.90	0.94	0.91	0.97
	45	0.83	0.90	0.86	0.96

**Table 2 entropy-22-01225-t002:** The prediction results with MMPE and Origin indicators on the Voldemort dataset.

Performance Indicators	Anomaly Detection Algorithms	Precision	Recall	f1-Score	Accuracy
**Origin**	OC-SVM	0.72	0.72	0.72	0.80
	Isolated Forest	0.89	0.83	0.86	0.96
**MMPE**	OC-SVM	0.86	0.86	0.86	0.82
	Isolated Forest	0.90	0.94	0.91	0.97

**Table 3 entropy-22-01225-t003:** The prediction results with MMPE and Origin indicators on the Google Cloud Trace (GCD) dataset.

Performance Indicators	Anomaly Detection Algorithms	Precision	Recall	f1-Score	Accuracy
**Origin**	OC-SVM	0.73	0.73	0.73	0.99
	Isolated Forest	0.88	0.83	0.84	0.99
**MMPE**	OC-SVM	0.73	1	0.85	0.99
	Isolated Forest	0.96	0.93	0.93	0.90

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
