# Peer review of "A Dynamic Anomaly Detection Approach Based on Permutation Entropy for Predicting Aging-Related Failures"

_entropy, 2020, doi:10.3390/e22111225_

Round 1

Reviewer 1 Report

The paper is very interesting work and show the problem of "software aging". Even well-designed software systems suffer from chronic performance degradation and this affects the security of the entire IT system. Estimating the place and time of failure is very troublesome. The authors clearly defined the purpose of the study and the presented results confirm the correctness of the assumptions.

The results are presented in the form of tables and graphs, which facilitates analysis and interpretation. Until now, the problem of "software aging" detection has been the insufficient aging indicators.

In my opinion authors sholud be a clearly discribe few aspects of their solution. Questions to the authors:

Can the types of errors be distinguished from the presented analysis and results: e.g. due to internal (e.g. software bugs) or external (e.g. resource exhaustion) constraints? If so, can they indicate critical points on the chart?

How much impact do the applied software monitoring systems have on the results?

Is it possible to estimate the Ahead-Time-To-Failure predictor? Can such a predictor be estimated for the data presented in the article?

Has the MMPE method been effectively estimated only analitical (or experimentally) in relation to the permutation entropy (PE) and the multi-scale permutation entropy (MSPE) and another multi-dimensional multi-scale entropy: multi-dimensional multi-scale entropy (MMSE). Has any attempt been made to determine which classifier is best: decision tree, nearest neighbor K, discriminant analysis, support vector machine?

How was the process validation on synthetic data.

Reviewer 2 Report

This is an interesting study but needs an extensive efforts to be considered for publication

Table 1 does not have any information and it is just some basic information. What are you trying to say? Instead of the description you need to fill the table with your real results

In Table 2, explain why F1-score is needed in addition to precision and recall. Please justify?

What are you trying to say by presenting figure 7, you already presented Table 2?

The related mathematical equations should be presented for isolated forest, and all other algorithms, and discussion should be added why you chose those algorithms, especially for your case study. In new section of method

A new section at the end of your manuscript should be added regarding applicability of the method for real life problem

Your figures are very confusing and unnecessary. If you present the results by table you do not need to present them again through figures.

Why you presented the codes like this below line 175. Organize the table, name it and refer it in the content of your manuscript, also find a better way of presenting the codes

Figure 5, around the solid line there is some area like confidence interval, what is that? Define it

Add problem statement, objectives and study contribution in the introduction section separately

Round 2

Reviewer 2 Report

My concerns have been addressed